# Validation of web-based audiometry version of HEARZAP

**Pandi Renganath P., Vidya Ramkumar** [ID]*

Faculty of Audiology and Speech Language Pathology, Sri Ramachandra Institute of Higher Education and Research, Chennai, Tamil Nadu, India

* vidya.ramkumar@sriramachandra.edu.in

## Abstract

### Aim

The purpose of this study was to verify the accuracy of the web-based audiometer HEAR-ZAP in determining hearing thresholds for both air and bone conduction.

### Method

Using a cross-sectional validation design, the web-based audiometer was compared to a gold standard audiometer. Participants in the study totaled 50 (100 ears), of which 25 (50 ears) had normal hearing sensitivity and 25 (50 ears) had various types and degrees of hearing loss. All subjects underwent pure tone audiometry, including air and bone conduction thresholds, using the web-based and gold standard audiometers in a random order. A pause between the two tests was allowed if the patient felt comfortable. The testing for the web-based audiometer and gold standard audiometer was done by two different audiologists with similar qualifications in order to eliminate tester bias. Both the procedures were performed in a sound treated room.

### Results

For air conduction thresholds and bone conduction thresholds, respectively, the mean discrepancies between the web-based audiometer and the gold standard audiometer were 1.22 dB HL (SD = 4.61) and 0.8 dB HL (SD = 4.1). The ICC for air conduction thresholds between the two techniques was 0.94 and for the bone conduction thresholds was 0.91. The Bland Altman plots likewise indicated excellent reliability between the two measurements, with the mean difference between the HEARZAP and the gold standard audiometry falling within the top and lower limits of agreement.

### Conclusion

The web-based audiometry version of HEARZAP produced precise findings for hearing thresholds that were comparable to those obtained from an established gold standard audiometer. HEARZAP, has the potential to support multi-clinic functionality and enhance service access.

**Data Availability Statement:** We have uploaded the data set in the Open Science Framework repository and it can be accessed through the link http://osf.io/x3svw

**Funding:** The corresponding author received in-part funding support from Izameditechnologies PVT Ltd. for this study. The testing lab and equipment used for the study was funded through the corresponding author's grant 'The Wellcome Trust DBT India Alliance grant' (IA/CPHI/19/1/504614). The funders did not play any role in the study design, data collection and analysis, decision to publish, or preparation of the manuscript.

**Competing interests:** The authors have declared that no competing interests exists

## Introduction

Globally, it is estimated that 466 million individuals suffer from hearing loss that is debilitating. The number of persons with hearing problems is rapidly increasing as the world population grows and ages. At least 700 million individuals will need hearing rehabilitation by 2050, when it is predicted that approximately 2.5 billion people would have some degree of hearing loss [1]. People with hearing loss who reside in low-resource regions are the least likely to obtain the resources they require to help them manage their impairment (e.g. hearing screening, hearing aids, cochlear implants) [2]. Therefore, management of hearing loss and their participation in ongoing care is delayed.

Tele-audiology, which involves remote ear and hearing testing, counseling and rehabilitation, has been explored for close to two decades. Opportunities for tele-audiology are growing as a result of technological advancements, and audiologists have adapted clinical tests to support remote service delivery to enhance service accessibility and better personalize services for patients and their families [3].

Telehealth initiatives are typically justified on the basis of reduced patient expenses due to averted costs related to travel, wage loss, accommodation [4–6]. However, telehealth can further accelerate access when the need for expensive sound booths and desktop based audiometric equipment is minimized to reduce provider's cost involved in remote hearing care.

In the last decade, innovations around audiometers have been undertaken to move away from desktop based systems to a more portable system in order to promote wider accessibility for hearing care without the need for expensive equipment and sound treated rooms [7]. A computer-based diagnostic audiometer, KUDUwave was developed and validated for tele- and booth-less audiometry [8]. Subsequently, there have been several explorations in software app-based audiometry especially for screening [9–12]. Further, software based diagnostic audiometry have also been explored with minimal hardware requirements that can be performed on laptops/ computers with internet connectivity [9, 13]. However, there are longevity and storage related limitations in software application-based systems. The app file is downloaded to the device, requiring a few permissions and a limited amount of storage space, and when an operating system is updated, it impacts the software application's ability to be used indefinitely. Hence, the program must be updated in accordance with the new permissions and compatibility. This increases the recurring development cost considerably.

Web-based solutions have the potential to overcome some of these challenges due to its interoperability, data access controls and privacy. Cloud/online operations are used in web-based audiometry and hence there is no storage concern due to file downloaded to the system or device. In addition, web-based solutions are compatible with a wide range of hardware and software platforms. They also require little maintenance and are simpler to update [12].

Web-based server applications to support remote audiometry in multiple clinics have been explored [12] but is limited to patient management and remotely access of a software compatible audiometer connected to the internet. More recently a speech audiometry web application was developed where children assist a virtual robot in assigning words to visuals [14] and also a web based diagnostic audiometer that relies on the digital-to-analog converter built into the transducer for desired output [15].

In order to support the implementation of a hub-and-spoke clinical practice model in semi-urban and semi-rural parts of India, a hybrid web plus app-based audiology solution HEARZAP (Iza Medi Technologies Pvt. Ltd.), was developed as a comprehensive solution to address screening, hearing assessment and hearing aid fitting needs for adults.

HEARZAP includes key modules such as, Quick hearing test, Advanced hearing test, Finding the right hearing aids and the e-Store (shop hearing aids/accessories) to promote delivery of hearing care at patient's doorstep. The *Quick Hearing Test* is a DIY 3-minute hearing screening test that uses a smartphone app to screen for hearing loss with an in-built noise monitoring mechanism. The *Advanced Hearing Test* is the diagnostic module to be performed by audiologists at a patient's home or in the clinic. *Finding the right hearing aids* is a module where artificial intelligence (AI) or machine learning (ML) based personalized recommendation engine considers the person's needs and lifestyle to suggest a suitable hearing aid. This suggestion will trigger a request to a certified audiologist, who will then connect with the patient to provide appropriate counseling and guidance based on the audiogram and the hearing aids suggested through the app and ensure suitability of the hearing aid for the patient. To close the loop from screening to rehabilitation, the HEARZAP also has an *e-Store (shop hearing aids/accessories)*, where individuals can then buy the recommended hearing aids, accessories, assistive listening devices, and ear protection devices from a range of manufacturers. The web-based software is compatible with both windows-OS and iOS and has 256-bit encryption for privacy protection.

While HEARZAP has three modules, this study was undertaken to validate the diagnostic audiometry having AC, BC & NBN masker module "Pro Test".

## Materials and methods

This study was approved by the appropriate Institutional Ethics Committee of Sri Ramachandra Institute of Higher Education and Research (Deemed to be University) (CSP/21/NOV/102/576). Informed written consent was obtained from all participants prior to participation in the study.

### Study design

Cross-sectional validation design.

### Sample

The sample size was estimated as 96 ears using the formula N = $Z^2$ PQ / L2, where Z = 1.96 (95% confidence level), P = Population proportion (90%), Q = 1-P and L is the allowable error (6%).

### Procedure

**Gold standard audiometry.** Pure tone audiometry (PTA) was conducted using a gold standard audiometer (HARP Inventis), calibrated as per ANSI S3.6 and IEC-60645-1 standards (ANSI, 2004; IEC, 2017). The testing was done using supra-aural headphones (TDH-39) and bone vibrator (RADIOEAR B-71).

An audiologist conducted a thorough case history, followed by an otoscopic examination. The evaluation began with the right ear and then in the left ear. Before beginning the audiometric testing, the participant was seated in a position that promoted safety and comfort while also allowing for accurate and valid assessment. The participants were instructed to press the response button whenever the tone was heard through headphones. The test instructions were given in a language or manner that the participant could understand. Inadvertent visual signals to the participant were reduced, and it was made easier to observe participant responses to stimuli. The responses were also monitored and reinforced. The duration of the pure-tone

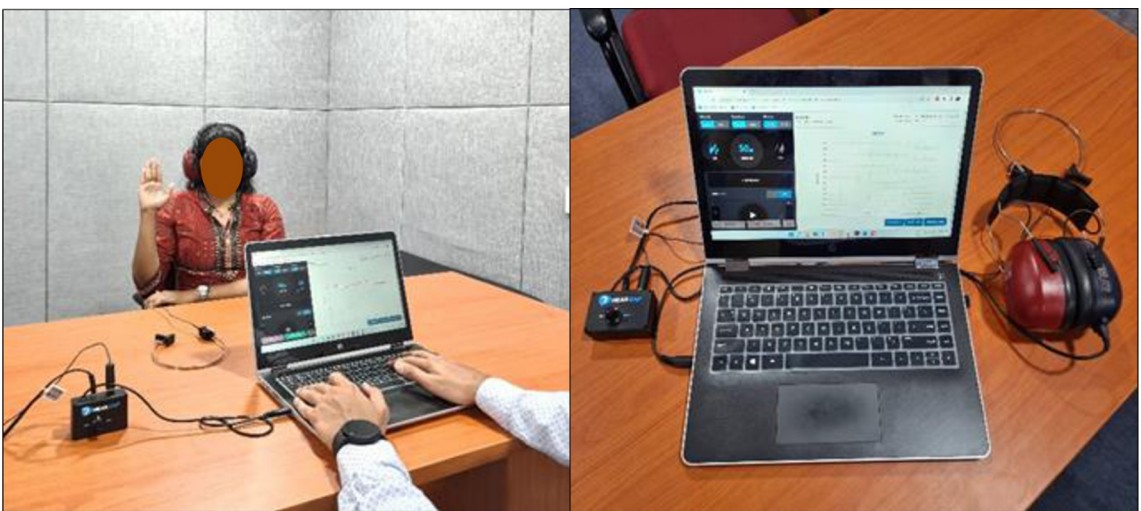

**Fig 1. web-based audiometry testing on a laptop connected to transducers via a splitter.** HEARZAP

stimuli presentation was for 1 to 2 seconds [16]. The time between each tone presentation varied, but it was never less than the test tone.

Air and bone conduction thresholds were obtained for frequencies 250Hz, 500Hz, 750Hz, 1kHz, 1.5kHz, 2kHz, 3kHz, 4kHz and only air conduction thresholds were obtained for 6kHz & 8kHz. The audiometric threshold was recorded for each frequency at the point when the participant responded to the signal twice using the modified Hughson-Westlake procedure [17] and there was no subjective response when the signal's intensity was lowered further. On the mastoid, a standard bone-conduction vibrator was positioned with the appropriate force applied. When masking was applied, the contralateral ear was covered.

To ensure uniform approach for both the gold standard and web-based audiometric testing, masking signals were provided to the non-test ear when appropriate in line with the masking plateau method [18].

**HEARZAP web-based audiometry.** *Device setup and calibration.* A commercial laptop with a stable internet connection was connected to the transducers using an external sound card and a splitter switch (HEARZAP connector) (see Fig 1). An online web-link (HEARZAP) was used to enter the web-based audiometry software. The output from the web-based audiometer from RADIOEAR B71 bone vibrator and RADIOEAR DD65v2 circumaural headphones were calibrated as per the ANSI S3.6 and IEC-60645-1 standards (ANSI, 2004; IEC, 2017). The maximum output in dB HL for all the test frequencies (Table 1), linearity, crosstalk and harmonic distortions were evaluated using Larson Davis Calibration kit (Model 824).

*Testing procedure.* Using a web-link, an audiologist conducted the diagnostic hearing testing (PTA). Air and bone conduction thresholds were obtained for frequencies 250Hz, 500Hz, 750Hz, 1kHz, 1.5kHz, 2kHz, 3kHz, 4kHz and only air conduction thresholds were obtained for 6kHz & 8kHz. The participants were instructed to raise their hand or finger whenever a stimulus was heard. Fig 1 shows HEARZAP web-based audiometry testing on a subject. The thresholds were recorded across frequencies mentioned above for both air and bone conduction using the same method followed for gold standard audiometry. The order of the test was randomized to minimize the order effect. A time gap comfortable to the patient was given between the two tests. The results of web-based audiometry were stored in the audiogram format in the HEARZAP cloud-based storage.

**Table 1. Maximum output value (dB HL) for air and bone conduction across frequencies using an external sound card.**

| Frequency (Hz) | Maximum Output Value (dB HL) | |
|---|---|---|
| | Air Conduction | Bone Conduction |
| 250 | 95 | 40 |
| 500 | 110 | 55 |
| 750 | 110 | 70 |
| 1000 | 110 | 75 |
| 1500 | 115 | 80 |
| 2000 | 110 | 80 |
| 3000 | 110 | 80 |
| 4000 | 105 | 75 |
| 6000 | 95 | - |
| 8000 | 90 | - |

The testing in web-based audiometer and gold standard audiometer were performed by two different audiologists with equal competencies in a random order. This was done to eliminate any tester bias.

## Analysis

Mean difference in thresholds obtained from gold standard audiometer and web-based audiometer were calculated for both air and conduction across frequencies. The Intraclass Correlation Coefficient was calculated to assess the reliability and absolute agreement between thresholds obtained in the gold standard audiometer versus the web-based audiometer. The extent of correlation was ascertained using the categories given by Koo & Li, (2016) [19]. The Bland Altman plot analysis was used to assess if there was any bias between the mean differences and also estimate the agreement interval [20]. The Statistical Package for Social Sciences (SPSS) version 16.0 was used for all of the aforementioned analysis.

## Results

In order to demonstrate the accuracy with which the HEARZAP audiometer can distinguish between ears with normal hearing and hearing loss, participants with both normal hearing sensitivity and those with hearing loss were examined. Fifty participants (24 males and 26 females) were recruited for the study from the outpatient clinic of a private medical college using a convenient sampling method. Of them, 25 participants had normal hearing sensitivity (50 ears) and 25 participants had varying degrees and types of hearing loss (50 ears). All participants chosen were between 18 and 65 years of age with a mean age of 38.5 years (SD = 13.2).

**Table 2. Number of ears under each degree of hearing loss.**

| Degree of hearing loss | No. of ears |
|---|---|
| Normal hearing | 50 |
| Minimal hearing loss | 23 |
| Mild hearing loss | 17 |
| Moderate hearing loss | 4 |
| Moderately severe hearing loss | 3 |
| Severe hearing loss | 3 |

**Table 3. Intraclass correlation coefficient for air conduction thresholds between HARP Inventis and HEARZAP audiometer (consistency and absolute agreement).**

| Frequency | AC | | BC | |
|---|---|---|---|---|
| | ICC for consistency | ICC for absolute agreement | ICC for consistency | ICC for absolute agreement |
| 250 Hz | 0.9 | 0.87 | 0.8 | 0.8 |
| 500 Hz | 0.9 | 0.89 | 0.87 | 0.86 |
| 750 Hz | 0.93 | 0.93 | 0.94 | 0.94 |
| 1 kHz | 0.92 | 0.92 | 0.88 | 0.88 |
| 1.5 kHz | 0.96 | 0.96 | 0.96 | 0.96 |
| 2 kHz | 0.94 | 0.94 | 0.91 | 0.89 |
| 3 kHz | 0.97 | 0.97 | 0.98 | 0.97 |
| 4 kHz | 0.94 | 0.93 | 0.9 | 0.88 |
| 6 kHz | 0.98 | 0.98 | - | - |
| 8 kHz | 0.95 | 0.95 | - | - |

Participants had mild to severe degree of Hearing Loss (Table 2), and included conductive (n = 12), sensorineural (n = 31), and mixed (n = 7) types of loss. The type and degree of hearing loss obtained was equivalent between the gold standard audiometer and the HEARZAP web-based audiometer, except in three ears with normal hearing. In these three ears, minimal hearing loss was obtained in HEARZAP audiometer.

Intraclass correlation coefficient was used to analyze the threshold data for air and bone-conduction tests using both gold standard and web-based audiometer to determine how closely two audiometers' thresholds resembled one another. The Intraclass correlation coefficient (ICC) for air conduction thresholds across frequencies tested shows a value of 0.94 (94% consistency) between the conventional and web-based audiometry. For the bone conduction thresholds, the ICC value was found to be 0.91 (91% consistency) across tested frequencies between the two methods.

The repeatability based on the correlation between recordings (consistency) and the repeatability based on the exact same scores (absolute agreement) was measured. Intraclass Correlation Coefficient for AC and BC thresholds (Table 3) are suggestive of an 'excellent reliability' as per classification given by Koo & Li (2016).

Web-based audiometer (HEARZAP) and gold standard audiometer had a mean difference in air conduction thresholds of 1.22 dB HL (SD = 4.61) (Table 4). The mean difference between bone conduction threshold was 0.8 dB HL (SD = 4.1) (Table 4). The majority of air and bone conduction thresholds from the web-based audiometer were within 10 dB HL of gold standard audiometry values, indicating clinical equivalency.

The Bland Altman plot was generated for threshold differences in AC (Fig 2A–2K) and BC (Fig 3A–3H) across frequencies. The green line indicates the mean and the red lines indicate the upper and lower limits of agreement (95% confidence interval). The plots demonstrate that, the majority of the data fit within the lower and upper limits of agreement closer to the mean. This implies that the majority of the thresholds identified using gold standard audiometer and web-based audiometer are consistent.

**Table 4. Mean difference of air and bone conduction thresholds (dB HL) between gold standard (GS) and HEARZAP (HZ) audiometer.**

| Frequencies (Hz) | | 250 | 500 | 750 | 1000 | 1500 | 2000 | 3000 | 4000 | 6000 | 8000 |
|---|---|---|---|---|---|---|---|---|---|---|---|
| Mean difference in dB HL | AC | 2.7 | 1.75 | 0.8 | 0.8 | 0.9 | 1.05 | 0.65 | 2.3 | 0.49 | 0.85 |
| Mean difference in dB HL | BC | 0.4 | 0.65 | 0.3 | 1.05 | 0.25 | 2.6 | -0.6 | 2.4 | | |

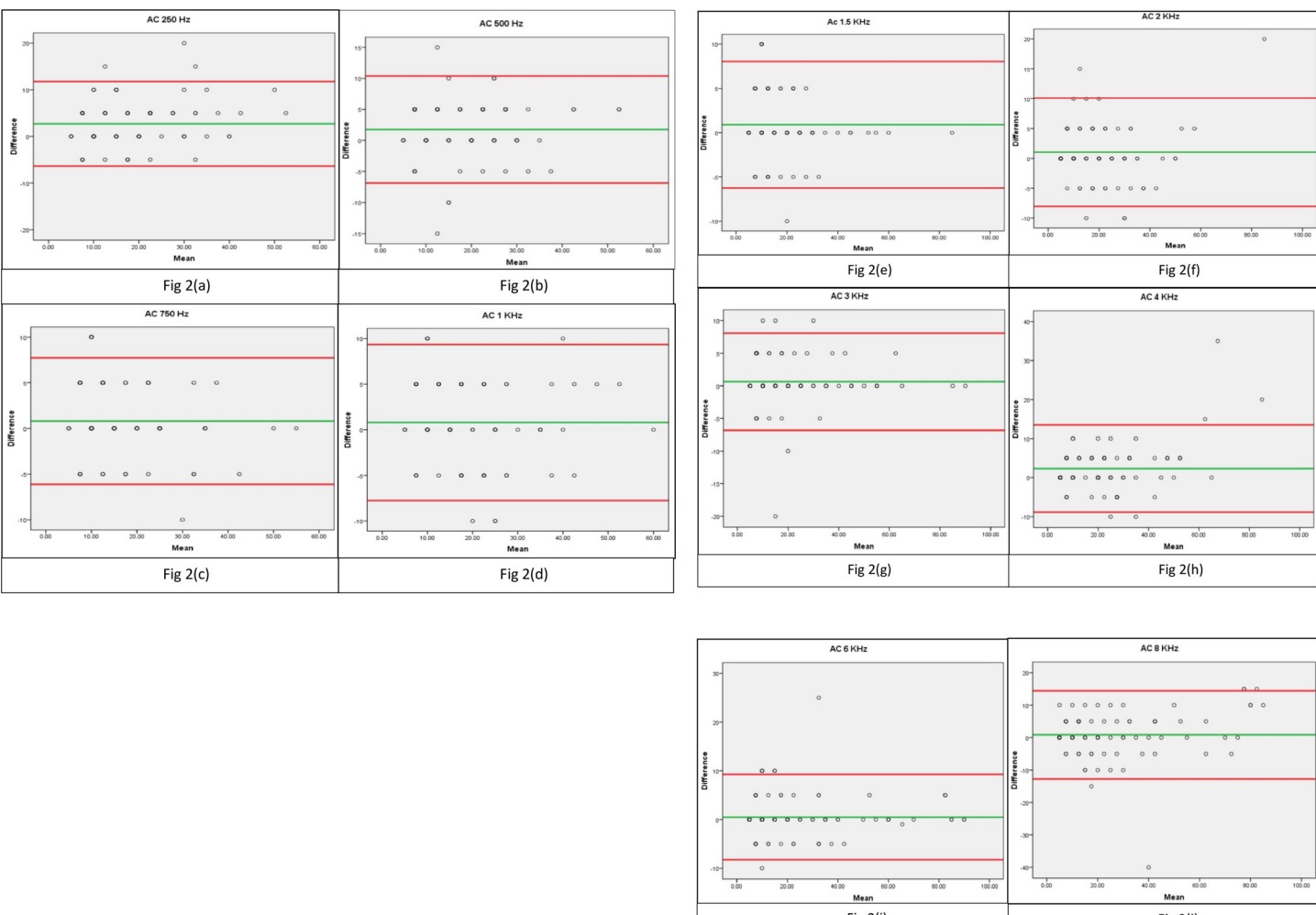

**Fig 2. (A-K): Bland Altman plots showing difference against mean for air conduction thresholds across frequencies between gold standard and HEARZAP audiometry.**

## Discussion

The findings of the study demonstrate clinical equivalence between HEARZAP web-based audiometer and the gold standard conventional audiometer in determining type and degree of hearing loss.

There have been several innovations and advancements in audiometers, including computer based [21], software apps [10, 11] and web-based systems [15]. These innovations have emerged to decentralize basic hearing assessment, especially for adults and geriatrics, from sound treated booths in clinics to the community site or homes.

The HEARZAP is designed as a comprehensive hybrid hearing care eco-system which includes a clinic console, management console and a mobile app. The clinic console includes a web-based audiometer as well as tools for arranging hearing test appointments, hearing aid trials, fine-tunings, returns, and service requests. Administrators can utilise the management interface to build sections or make modifications to the clinic console. Patients or the general public can use the mobile App to conduct a hearing screening from the comfort of their home.

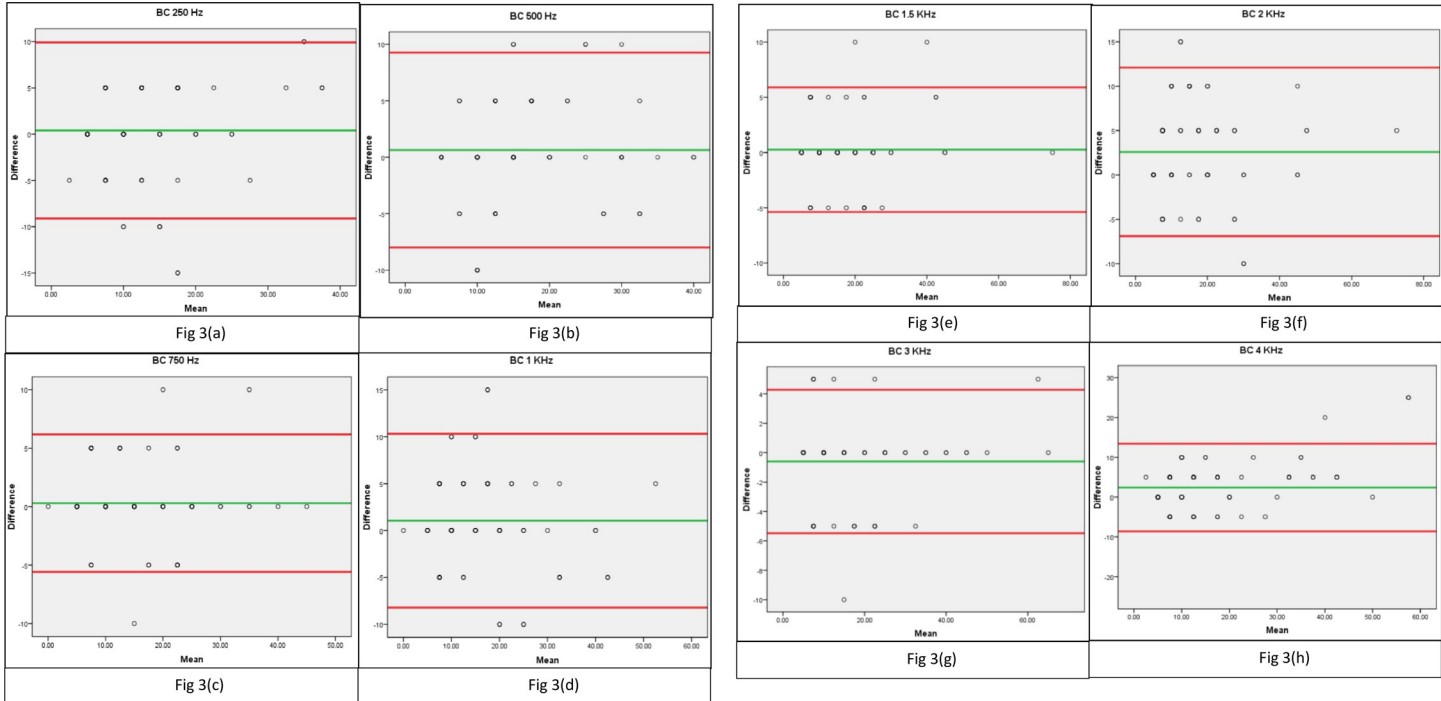

**Fig 3. (A-H): Bland Altman plots showing difference against mean for bone conduction thresholds across frequencies between gold standard and HEARZAP audiometry.**

Such a comprehensive solution is expected to be more acceptable and suitable to the clinical practitioners [12].

In general, research involving validation of computer, smartphone, and web-based audiometers have included a lesser proportion of persons with hearing loss [12, 21], except one study [15] in which 95% participants had hearing loss, but they did not account for the various types of loss. In the present study 50% of the participants had varying degrees of hearing loss and was also inclusive of all types of hearing loss. A balanced inclusion of participants with hearing loss is useful to capture reliability and accuracy measures more precisely.

The majority of the ears analysed (97%) by the HEARZAP audiometer exhibited similar degrees of hearing loss in comparison to the gold standard conventional audiometer. Notably, the HEARZAP audiometer detected a minimal degree of hearing loss in three of the hundred ears examined (3%) whereas the gold standard audiometer detected normal hearing sensitivity in accordance with Goodman's classification [22]. This may be due to factors such as participant attention or response tasks.

The mean difference in air and bone conduction thresholds found in our investigation was similar to that obtained in computer-based audiometry [21], with the exception of a slightly better mean difference in bone conduction obtained by Thai-Van et al. (2022) in their web-based audiometry.

Bland Altman plot was used in addition to Intraclass Correlation Coefficient which indicated similar results to the findings of an automated smartphone app that established bone-conduction pure-tone thresholds [23]. The majority of plots resulting from the mean difference between the gold standard and HEARZAP audiometer lie between the upper and lower limits of agreement (+/- 1.96SD). The clinical equivalence of the HEARZAP audiometer and the gold standard audiometer is thus comparable to the findings from earlier studies.

The HEARZAP audiometer can be employed as an effective option to minimize the need for hardware-based test equipment for hub-and-spoke or multi-center tele-audiology clinics. Yet, the HEARZAP audiometer's current limitation is the lack of a noise monitoring system included into the web-based software. However, these enhancements are planned in the future design.

## Conclusion

HEARZAP is the first known web-based comprehensive hearing care solution including hearing screening, hearing threshold assessments, prescription of hearing aids based on hearing loss, and enabled with purchase portal, all under supervision of a certified audiologist. Indigenous solutions such as HEARZAP, are likely to reduce costs and promote wider utility of such solutions. The reliance on bulky desktop-based audiometric equipment can be reduced with HEARZAP web-based audiometer to enhance service access.

## Acknowledgments

Mr. Rajapandian S, Founder, Iza Medi Technologies Pvt. Ltd, who has developed the HEARZAP technology and partnered with us for this research.

## Author Contributions

**Conceptualization:** Vidya Ramkumar.

**Data curation:** Pandi Renganath P.

**Formal analysis:** Pandi Renganath P.

**Funding acquisition:** Vidya Ramkumar.

**Investigation:** Pandi Renganath P., Vidya Ramkumar.

**Methodology:** Vidya Ramkumar.

**Project administration:** Vidya Ramkumar.

**Resources:** Vidya Ramkumar.

**Software:** Vidya Ramkumar.

**Supervision:** Vidya Ramkumar.

**Validation:** Vidya Ramkumar.

**Visualization:** Vidya Ramkumar.

**Writing – original draft:** Pandi Renganath P.

**Writing – review & editing:** Vidya Ramkumar.

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
