## [Decision Letter · Decision Letter 0]

17 Feb 2023

PONE-D-22-32800Validation of web-based audiometry version of HEARZAPPLOS ONE

Dear Dr. Ramkumar, 

Thank you for submitting your manuscript to PLOS ONE. After careful consideration, we feel that it has merit but does not fully meet PLOS ONE’s publication criteria as it currently stands. Therefore, we invite you to submit a revised version of the manuscript that addresses the points raised during the review process.

The work is interesting, but need some minor revisions. 

We look forward to receiving your revised manuscript.

Kind regards,

Annalisa Pace

Academic Editor

PLOS ONE

Journal Requirements:

Journal Requirements:

Reviewers' comments:

Reviewer's Responses to Questions

**Comments to the Author**

1. Is the manuscript technically sound, and do the data support the conclusions?

Reviewer #1: Yes

2. Has the statistical analysis been performed appropriately and rigorously? 

Reviewer #1: Yes

3. Have the authors made all data underlying the findings in their manuscript fully available?

Reviewer #1: Yes

4. Is the manuscript presented in an intelligible fashion and written in standard English?

Reviewer #1: No

5. Review Comments to the Author

Reviewer #1: The authors propose the validation of a web –based audiometry system and so the introduction of it onto the clinical practice: a good solution for hearing screening, identification of thresholds, also for aids prescription and management, reducing costs and enhancing service access to all patients.

Criticism: - Lacking word and error in abstract; - Please write “Figure 1” as “Fig 1” and so on; - Please insert Fig 1 citation in the text; - Please correct extra space in the text; - Please cite references in brackets in the text.

6. PLOS authors have the option to publish the peer review history of their article (what does this mean?). If published, this will include your full peer review and any attached files.

Reviewer #1: No

---

## [Author Response · Author response to Decision Letter 0]

21 Feb 2023

A point by point response to reviewer is provided in the files uploaded for the perusal of the reviewer

---

## [Editor Report · Decision Letter 1]

13 Mar 2023

Validation of web-based audiometry version of HEARZAP

PONE-D-22-32800R1

Dear Dr. Ramkumar,

We’re pleased to inform you that your manuscript has been judged scientifically suitable for publication and will be formally accepted for publication once it meets all outstanding technical requirements.

Kind regards,

Annalisa Pace

Academic Editor

PLOS ONE

---

## [Editor Report · Acceptance letter]

17 Mar 2023

PONE-D-22-32800R1 

Validation of web-based audiometry version of HEARZAP 

Dear Dr. Ramkumar:

I'm pleased to inform you that your manuscript has been deemed suitable for publication in PLOS ONE. Congratulations! Your manuscript is now with our production department. 

Kind regards, 

on behalf of

Dr. Annalisa Pace 

Academic Editor

PLOS ONE